# Talking about Oneself to Talk about Christ: The Autobiographical Text of Philippians 3:1–4.1 in Light of Ancient Rhetorical Heritage

Francesco Bianchini 

Theology Faculty, Urbaniana University, 00120 Rome, Italy; f.bianchini298@gmail.com

**Abstract:** In this contribution, we will proceed in three steps. First of all, we will investigate the rhetorical approach for studying the Pauline letters, considering different methodological options. In this context, we will propose the approach of the literary rhetoric as the most valid. Secondly, we will analyse the autobiographical text of Philippians 3:1–4:1, starting from its delimitation, textual criticism, and its arrangement, according to oral and discursive models. Then, we will proceed with genre and literary origins; here, we will discover the periautologia as the point of reference of the Pauline autobiography. This eulogy of self is a genre, well known in the rhetorical tradition, to which Plutarch dedicated the treatise *On praising oneself.* This discovery determines the following exegetical analysis of the text. Thirdly, we will conclude with a reflection about Paul's way of speaking about himself in this passage. In light of ancient rhetorical heritage, he does not use his autobiography to praise himself but to praise Christ, who completely changed his life. Ultimately, Paul's talk about himself is a way of talking about Christ for the benefit of the addressees who should creatively imitate the Apostle and his Christian life.

**Keywords:** Philippians 3; Pauline autobiography; Pauline Rhetoric; periautologia

## 1. Introduction

Within the Pauline corpus, the "Letter to the Philippians" is to be included among the lesser works of the Apostle with regard to its length (1629 words, corresponding to 104 verses, subdivided into 4 chapters in the printed editions). Nevertheless, such a classification does not appear adequate when judging its importance. First of all, because one of its texts, the Christological passage of 2:6–11, has represented a point of reference not only for theology but also for liturgy and Christian life over the course of centuries and to this very day. In addition, the importance of the letter emerges from those eminently personal passages such as chapter 3, where Paul, a prisoner on account of the Gospel and with the possibility of martyrdom before him, shows to the recipients and to subsequent readers, more than in any other of his writings, the profound and mysterious relationship which binds him to his Lord.

Therefore, in this contribution, we intend to analyse the autobiographical text of Philippians 3 in light of the ancient rhetorical heritage in order to understand in depth the way in which Paul talks about himself.[1] For this purpose, we will proceed in three steps. First of all, we will investigate the rhetorical approach for studying the Pauline letters, considering different methodological options. Secondly, in line with our choice to use a literary rhetoric approach, we will analyse the autobiographical text of Philippians 3:1–4:1, starting from its delimitation, textual criticism, and its arrangement. Then, we will proceed with the literary genre; here, we will discover the periautologia as the point of reference of the Pauline autobiography. This discovery determines the following exegetical analysis of the text. After this, we will put the text in the context of the letter in order to understand the overall logic of Philippians 3:1–4:1. Thirdly, we will conclude with a reflection about

Paul's way of speaking about himself in this passage and also draw a comparison to the talking about oneself that was developed and implemented in his cultural context.

## 2. Rhetorical Analysis of the Pauline Letters

Since the mid-1970s, the Pauline letters have been studied in a rhetorical perspective, due to the fact that scholars recognized their persuasive and argumentative character, and rhetoric was the art of persuasion *par excellence* in the first century AD.[2] Today, after almost 50 years, to speak of rhetorical analysis of the Pauline letters and of the NT can be quite vague; in fact, the interpreter needs to specify which kind of rhetorical analysis s/he intends to follow.

We can find at least four approaches currently available for those seeking to interpret the Pauline letters from a rhetorical lens: rhetorical criticism, biblical rhetoric, new rhetoric, and literary rhetoric.[3] The first approach makes use of the classical manuals of rhetoric, above all, the analysis of the Pauline letters with the conviction of the eminently persuasive tenor of the Apostle's writings. The second approach is described by its supporters as that of biblical rhetoric, set in contrast with the first, and it claims that all the biblical texts, whether Old Testament or New Testament, were composed according to a predetermined plan that retraces the particular symmetrical compositions of Semitic culture. The third approach contemporises the ancient rhetorical heritage, making use of the contributions of modern disciplines such as linguistics, semiotics, anthropology, and sociology; presenting itself as a real theory of persuasive discourse; and fixing its attention on the argumentation itself, classifying the different types of argument. The fourth and last approach, that of the literary rhetoric method of analysis, deriving from Aletti[4] and other French and Italian scholars, draws on Greco-Roman rhetoric as a tool for understanding the NT, especially Paul's letters. Such an approach avoids the rigidity arising from rhetorical criticism, in which rhetorical models (linked mainly to the judicial genre) become a straitjacket hindering the liberty of the expression of Paul and other NT authors. Literary rhetoric, in distinction, bypasses the purely formal level of earlier rhetorical analyses as a way to draw out the persuasive dimension of the text along with its performative function vis à vis the recipients. Using the text's composition and rhetorical figures, the aim of literary rhetoric is to elucidate the development of the text's argumentative flow, especially by analysing its relative proof, thereby uncovering the overall message contained therein. This method thus combines the purely literary dimension with the discursive, and epistolography with rhetoric, in order to overcome the harmful dichotomies arising from previous rhetorical criticism of Paul's letters.

For the aforementioned reasons, we propose following this last rhetorical approach. Moreover, the fruitfulness of this literary rhetorical approach will become clear as we proceed in the study of Philippians 3. In contrast with rhetorical criticism, following Longenecker's distinction in regard to rhetoric in Galatians (Longenecker 1990, p. cix), the method of literary rhetoric is not a "diachronic rhetorical criticism", but rather a "synchronic rhetorical criticism", bestowing primacy upon the text itself rather than upon various comparisons with ancient rhetorical models. An assessment of the concrete results yielded from such an approach will determine the appropriateness of its application for Paul's corpus.

## 3. Exegesis of Philippians 3:1–4:1[5]

### 3.1. Delimitation

The first question to be posed is where the text we wish to examine begins and ends. On this level, it is not enough to refer to the numeration of the Bible in chapters and verses, which do not belong to the original text but were introduced in the 13th and 14th centuries.

With regard to the higher limit, we should note that there is a break between the end of chapter 2 and the beginning of chapter 3. In fact, there is a move from the third person singular with which Epaphroditus is mentioned (2:30) to the second person plural of the imperative for addressing the listeners directly, to put them on their guard against

adversaries (3:1–2). However, the question appears more complicated if we observe the sudden change in tone from exhortation to rejoicing (verse 1a) to polemical attack (verse 2); moreover, it lacks a syntactical connection. These observations have led several scholars to assume, especially in the past, the presence of two different letters clumsily joined here.[6] Nevertheless, if, on the one hand, the break between verse 1 and verse 2 is undeniable, on the other, verse 1b, as a reflection on the writing itself ("writing the same things"), acts as a good introduction to everything that follows, particularly to the following verse, which aims to capture the recipients' attention at the beginning of a new epistolary development. On account of what we have shown, therefore, we consider 3:1 the higher limit of the pericope of chapter 3.

If we seek the lower limit of the passage, we note the presence of the conjunction ὥστε, which has an inferential value so as to connect 4:1 with what is written before. This solution, which leads to the integration of the first verse of chapter 4 with our passage, is further supported by the parallelism of expression, almost an inclusion, between 3:1 ("my brothers, rejoice in the Lord") and 4:1 ("my brothers, stand firm in the Lord"). In conclusion, we can delimit our field of study in Phil 3:1–4:1 with a fair degree of certainty.

### 3.2. Textual Criticism

Seeing that we do not possess the manuscript that Paul dictated, textual criticism seeks to arrive at a reconstruction of the text that is reasonably close to the original, critically analysing the variants that have come down to us. In our passage, we did not have many textual problems. We could only indicate that the 28th edition of the *Novum Testamentum Graece* of Nestle–Aland displays uncertainties with regard to the insertion or omission of the following words in Phil 3:1- 4:1: ἀλλ' (verse 7); τήν (verse10); τῶν (verse 10); Ἰησοῦ (verse 12). However, the inclusion or not of these terms does not change the sense of the text.

### 3.3. Arrangement According to Oral and Discursive Models

Understanding how the passage is structured is a fundamental key to arriving at its correct interpretation. The oral model employs literary criteria (grammar and syntax, repetition of words and themes) to delineate the basic composition of a text, which must be perceived immediately by the listener in its essential nature. For the oral model, we can propose the following arrangement for our passage:

3:1 transition;
A. 3:2–4a comparison "we"/"they", with communication "I"–"you";
B. 3,4b–16 auto-presentation of "I", with link to "we";
A'. 3:17–21 comparison "we"/"they", with communication "I"–"you";
4:1 conclusion.

This scheme is useful because it provides us with a subdivision that links two units on account of intratextual echoes (A.–A'.) and highlights the special nature of verses 3:4b–16 (B.). Furthermore, we should note that the outer units consist of exhortations that are based on the exemplary model depicted in the central unit; on the other hand, the exhortations tell us about the perspective within which the exemplary model is to be read.

On the other hand, the discursive model makes use of rhetorical criteria to demonstrate the logical arrangement of the text, with reference to the typical disposition of the rhetorical discourse (dispositio) but also to its figures (elocutio). In the past, some exegetes (e.g., Harnisch 1999; Edart 2002; Marguerat 2004) have sought to apply this model to Phil 3:1–4:1. However, our pericope does not have a markedly argumentative character, given that Paul does not intend to prove a concept or a theory but, rather, to show his way of life. Consequently, the discursive model is shown to be unsuited to the nature of the passage, and, therefore, we follow the aforementioned oral model. Nevertheless, the broader rhetorical perspective of the study, whereby we seek to understand how a Pauline text proceeds and develops in order to engage and convince the audience, will be of great use to us and will be carefully considered by us in the subsequent analysis.

*3.4. Literary Genre*

Seeking the genre means comparing the passage with the various literary models of the time of Paul. The most widespread proposal among scholars has been that regarding the classical literary form of the exemplum. In it, a person or an event are offered as models of reference for the edification of the audience. Thus, in our text, Paul becomes an example of the Christian life presented in such a way that the listeners imitate him in their actions (cf. verse 17).[7] This identification with the exemplum appears to be well grounded. However, it does not turn out to be a total match for the complexity of Phil 3:1–4:1. In fact, the typical mode of expression of this literary genre is in the third person, whereas an important portion of this pericope is in the first-person singular (verses 4b-14). Additionally, the presentation which the Apostle makes of himself appears to be marked by encomiastic elements.

Thus, the most recent idea is that of identifying the literary genre of the passage as that of the periautologia, or eulogy of the self.[8] The Greek word περιαυτολογία is used for the first time by Plutarch in *On Praising Oneself* (about 100 AD), a section of his *The Morals*,[9] coming from the verb περιαυτολογέω, which means "to talk about oneself", understood in a positive sense. Plutarch's treatise is post-Paul, but it captures a rhetorical practice that has been widespread for centuries (and often frowned upon) and then fixed in a literary genre that has many points of contact with that of the autobiography (Cf. Pernot 1998). As Forbes (1986, p. 8) states, in the periautologia, typical features or *topoi* of the eulogy, which were pronounced in the first person, are adapted to the first person. In general, they comprise the following elements: origins, education, deeds, and virtues, with the possibility of also including the factor of comparison. However, in the ancient world, it is difficult to emphasise the individual; thus, the possibility of a recourse to the periautologia is found to lie mainly in two reasons: one is apologetic in nature, the other ethical. Following the former reason, it is appropriate to employ this form in order to defend oneself from the accusations of enemies, while on the basis of the second perspective, the eulogy of self has to constitute a means for imitating the author himself, suggested as a model of values and behaviour. In any case, the periautologia always turns out to be unpopular. Thus, according to the recommendations of the ancients, the orator must focus all his attention to lessen the unwelcome effect produced on the audience from a person who talks about himself by praising himself. In this context, therefore, there is recourse to a procedure that can be considered, following Pernot (1998, pp. 114–15)—a real occasion of transfer. In fact, if the periautologia has as a formula, "I am praising myself before an audience", the whole of the rhetorical skill consists of dissociating the "I" from the "me" or the orator from the audience. Thus, to disguise the "I", one puts one's own praise into the mouths of others; to disguise the "me", one reports personal merits to fortune or divinity or mixes one's own praise with that of the audience or others to whom one is related; to detach oneself from the audience, one presents periautologia in the form of an apostrophe towards opponents. In addition, little tricks that serve to attenuate and justify the praise of self are the blame of the conduct opposite to that of the one who is speaking and the list of some minor defects of the latter.

The above elements, typical of the periautologia, can be found with some certainty, in the passage of Phil 3:1–4:1. First of all, this form fully accounts for the employment, in verses 4b-14, of the first person as the subject of almost all the verbs; here is an "I" who is telling about his life in the past, present, and future. In this way, the Pauline text also appears to fulfil the necessary pre-condition of truthfulness of the periautologia[10], because it is based on a real autobiography. Next, in verses 5–6, we recognise some *topoi* from the encomiastic genre, such as origins (circumcision on the eighth day; of the people of Israel; of the tribe of Benjamin; Hebrew, born of Hebrews), education (living as a pharisee), and acts (persecutor of the Church, found blameless), based on his virtues (zeal and justice). In its turn, the rhetorical element of comparison between Paul and his enemies is directly present in verse 4, and in verses 2–3,18–21, is present indirectly, mediated by the group "we" to whom the Apostle belongs. If we dig still deeper into the reading of the text, we

also see that the two principal reasons for having recourse to the eulogy of self are present: in verses 2,18–19, the apologetic reason due to the hostile action of the enemies, and, in verse 17, the ethical reason bound up with the imitation of the good example of the subject who is praising himself. Additionally, the process of transfer in Phil 3:1–4:1 is brought into play on at least two levels. In fact, in verses 3,15, 20–21, the Apostle blends his own eulogy with that of the audience, and the one who is speaking shows himself the representative *par excellence* of the category that he is exalting, operating a transfer from himself to his listeners. In verses 7–8, on the other hand, Paul ascribes his own merits to the action of his Lord by performing a transfer of himself to Christ. Finally, in verses 12–14, the Apostle refers to his imperfection as a Christian, employing the trick of also citing his own limits so as to make his praise of himself more acceptable. In conclusion, seeing a periautologia with an exemplary function in this passage turns out to be the soundest proposal for understanding the text.

However, agreeing in this with Schmeller (2015), we note that a full identification of Phil 3:1–4:1 with this literary genre and its normal use in antiquity is not immediate, especially since verses 7–11 cannot be adequately explained. In fact, in these verses, unlike in the typical periautologia, Paul does not speak of the praiseworthy deeds he has done, but of the work accomplished in him by Christ; it is not his personal success that is narrated, but the losing of everything for the sake of a greater good; above all, the Apostle's "I" is not actually placed at the centre, but rather the person of Christ. Thus, on the one hand, the motive of boasting becomes paradoxical, consisting of a loss; on the other hand, the transfer process is implemented in a radical manner, as the identity of the Pauline "I" is completely transformed. At the end of this comparison, we believe that, on the one hand, the freedom with which Paul uses the literary canons of his time clearly emerges, and, on the other hand, the subsequent analysis of the text must be carried out, bearing in mind, above all, this paradoxical perspective of the periautologia of Phil 3:1–4:1.

### 3.5. Exegetical Analysis

Now that the time has come for an exegetical analysis, we shall ask about each of the expressions in the text. To this end, the use of words in Greek, in the LXX and the New Testament (especially in the Pauline passages), furnishes indications for grasping the sense that they take on in the specific context of the passage in question. Moreover, our interpretation will be guided precisely by what has emerged concerning the literary genre that determines the entire logic of our passage.

### 3.5.1. Transition (3,1)

The beginning of chapter 3:1, as mentioned, has a transitional function. In particular, verse 1a picks up the theme of joy, and the related exhortation, present in 2.18, occurs after an interruption due to autobiographical news and the recommendation of Timothy and Epaphroditus of 2:19–30, whereas verse 1b, also corroborated by the adverbial expression τὸ λοιπόν of verse 1a (cf. 1 Cor 1:16; 4:2; 7:29; 1 Thess 4:1; 2 Thess 3:1), introduces a new development of the writing, namely the second part of the letter, which is seen as a repetition of the first ("to write the same things"), plausibly at the level of content, argumentative tools, and exhortative purpose.[11]

### 3.5.2. Exhortation with Motivation (3,2-4a)

The textual unit A. (verses 2-4a), consists of a negative warning about the enemies (verse 2) and the related motivation given by the profile of the believers who place their trust in Christ (verses 3-4a). The exhortation in verse 2 is addressed to the recipients to beware of certain individuals, designated by insulting epithets.[12] In a manner typical of invective and used against opponents, Paul attacks them in order to discredit them in the eyes of the Philippians and prevent the former from exerting their evil influence on the latter (Cf. Neumann 1998). As for the identity of these adversaries, they are probably Judeo-Christians, who, according to the Apostle, constitute a possible danger to the ethno-

Christians of Philippi, since they would invite them to assume circumcision, prescribed by the Law,[13] and thus disavow the justification that comes solely from faith in Christ (3:9).[14]

In fact, verses 3-4a, providing the motivation for the exhortation of verse 2 (note γάρ), insist on the sign of Jewish identity, which is circumcision, a symbol of belonging to the Abrahamic covenant and a necessary element for participating in the temple liturgy and thus approaching God (cf. Ex 12:44–48; Ezek 44:7). In verse 3, Christians, especially those from paganism and uncircumcised, such as the Philippians, are now referred to as "the circumcision", in contrast to their opponents who represent "the mutilation". For the former, by virtue of the Spirit, worship the Lord through a life relationship with him (cf. Rom 12:1; Col 2:11). Their identifying mark is not circumcision, the more obvious element of "having confidence in the flesh" (cf. verse 5) but "boasting in Christ Jesus", placing the basis and reason for existence in Christ. All this is true, although Paul may also be confident in "the flesh" (verse 4a), i.e., on the gifts received and virtues acquired by him (cf. verses 5–6).

Taken together, these verses introduce the protagonists of the passage: Paul, Christ and the Philippians, and, in the background, the opponents. In this way, verses 2-4a are preparing Paul's self-eulogy, which will be developed from verse 4b. In fact, they set the opponents in play (so, too, do verses 18–19)—a typical reason for turning to the periautologia—and praise the group "we", preparing the eulogistic transfer from the author to his listeners and, finally, insert the rhetorical element of the comparison between Paul and his adversaries—correspondingly, between "trusting in the flesh" and "boasting in Christ Jesus".

3.5.3. Self-Eulogy with an Exhortatory Conclusion (3,4b-16)

The textual unit B. (verses 4b-16) brings Paul's "I" into greater prominence. We are before the periautologia proper, with the exclusive use of the first-person singular (verses 4b-14), followed by a paraenetic conclusion that is characterised by "we" (verses 15–16). This unit can be further divided, on the basis of grammatical and syntactical considerations, into four subunits: verses 4b–6; 7–11; 12–14; and 15–16. With regard to verses 4b–14, this arrangement is confirmed at the level of the literary genre since we are looking at a piece of self-boasting in three steps: Jewish boast (the past), boast *turned upside down* in Christ (from the past to the present), and moderated Christian boast (from the present to the future). Each stage is marked by the use of a verb from the semantic field of "think, consider", indicating three different moments in self-perception (δοκέω, verse 4b; ἡγέομαι, verses 7–8; λογίζομαι, verse 12).

Verse 4b announces the beginning of Paul's self-eulogy in response to the claims of a hypothetical exponent of a group of adversaries. In this way, the Apostle, while starting on the same level as the antagonist, also introduces a prodiorthosis, a rhetorical figure indicating a prior apology for something that may be offensive—in this case, for the periautologia he is about to weave. Hence also the rhetorical comparison, through which the author asserts that he has more reason than anyone else to place trust "in the flesh" as an alternative to trusting in Christ. Therefore, following the *topoi* of the encomiastic genre, verses 5–6 provide Paul's reasons for "trusting in the flesh", through a list of the privileges received (the first four) and the merits acquired (the other three). The privileges received are related to the encomiastic *topos* of origins and indicate that Paul was circumcised on the eighth day as an authentic Jew,[15] that he belongs ethnically to the people of Israel (hence not a proselyte), that he comes from the prestigious tribe of Benjamin (cf. Rom 11:1), and that his parents are both Jews. On the other hand, the merits acquired are related to the *topos* of education, because Paul was brought up within the Pharisaic current, the most rigorous as regards the practice of the Law (cf. Gal 1:14). Secondly, the merits acquired are related to the *topoi* of deeds and virtues, consisting of zeal for the Law, on account of which the Pharisee Saul persecuted the church, and righteousness resulting from legal observance for which he was blameless.[16]

In verses 4b–6, the Pauline autobiographical data are thus placed at the service of a self-praise that is presented through its seven elements, placed in a rhetorical *climax* (a figure in which successive words, phrases, clauses, or sentences are arranged in ascending order of importance) in order to constitute an impeccable and inimitable Jewish profile. In this way, verses 4b–6 also prepare the radical turn of verses 7–11, indicating that if Paul subsequently chose Christ, he did not do so to compensate for his failure in Judaism, but only because of the unexpected intervention of God, the only one capable of upsetting his firm and convinced personality.

Thus, with verses 7–8, a total reversal of the previous Jewish boasting is triggered. These verses enunciate, using a rhetoric of excess, that Paul has come to consider the "gains" from the excellent gifts and merits acquired in verses 5–6, "a loss", "rubbish".[17] Indeed, everything has now lost its value for him. Rhetorically speaking, we have an *anticlimax*, in full contrast to the *climax* of the previous verses. The reason for this revaluation and change is solely Christ, the encounter with and knowledge of the Risen One, who has become, for Paul, "my Lord". The concluding sentence of verse 8, "in order that I may gain Christ", attests to the fact that such knowledge is a dynamic and evolving reality. This sentence is deepened and clarified at the beginning of the next verse by "and be found in him"; it is not Paul who gains Christ, but it is Christ who causes him to be found in him.

As a whole, verses 9–11 show what derives from this reversal, what is now really important for Paul[18]. First of all, being united to Christ, with a state of justice before God, which is not based on the observance of the Law but on faith (verse 9). There are many questions of interpretation raised by verse 9, but it is essential to understand its syntactic structure correctly in order to provide a correct reading of it—"and be found in him, not having my own righteousness that comes from the law, but that [my own righteousness] which comes through faith in Christ, the righteousness from God based on faith". Thus, in the verse, the opposition is between "Law" and "faith in Christ" as two contrary and alternative principles on which to base one's righteousness, where only the last one leads to righteousness derived from God and is therefore salvific.[19] Paul, in his choice for Christ, opted for the second principle and abandoned the first.

The second effect resulting from the encounter with Christ is the experience of knowing him in the gradual, daily conformation to his death, which leads to the experiencing of the power of resurrection, even in the midst of suffering (verse 10). If, in verse 8, it was a matter of having come to know Christ, now, in the foreground, is the dynamic of knowing Christ that Paul lives day by day, reproducing, sustained by God's action, his own journey of death and resurrection (cf. 2.6–11).[20] In verse 10, the rhetorical figure of the *hysteron-proteron*, which reverses the natural order of events with the precedence of the element of Christ's resurrection over his sufferings, is intended to highlight that the Apostle, like every baptised person, first of all experiences the Lord with all his power as the Risen One (cf. verse 8) and then can and must also experience with him the tribulation, resulting from his own choice of faith. Moreover, the progressive conformation to Christ's death, during the time of earthly life, is marked by the hope, which does not depend on the person's will but on God's, of attaining the final resurrection[21] and thus full life (verse 11).

Thus, in verses 7–11, taken together, the Apostle performs a radical periautological transfer. In fact, here it is not simply a matter, as Plutarch also advises, of concealing self-praise by reporting some of one's merits to fortune or divinity; this is because his boast has been completely transferred to Christ and is motivated not by his successes but by what he has lost and fulfilled in him by the Lord. His self-eulogy has thus become a paradoxical one. Paul not only overturns his own Jewish eulogy but upends all the classical conventions of the periautologia, placing at the centre, not his "I" so much as the person of Christ.

Now, if verses 7–11 could lead us to suppose a completeness and perfection in the Apostle's existence and boast "in Christ", it is clear that what is said in verses 12–14 establishes a necessary clarification to avoid misunderstandings. Therefore, we have a double rhetorical *correctio*, the amending of the term or phrase just employed. Particularly, verse

12 corrects verses 7–11, saying that in following the path of the Christian life, Paul is not perfect and has not yet arrived, but he strives to reach the goal while Christ has laid hold of him. On the other hand, verses 13–14 correct verse 12, saying that Paul has not attained the destination of his journey, but he pursues the prize related to the high calling of God through Christ (that is, salvation in full and definitive communion with the Lord). In these last verses we find a metaphor that was much utilised also in the philosophico-moral teaching of the time, in relation to the struggle for virtues and ethical values.[22] Paul is a runner who does not look back at the course he has already completed but is wholly stretching forward towards the finishing post in order to gain the prize. Thus, the subunit of verses 12–14 is characterised by a toning down of Paul's Christian boast, which was presented with all his power in verses 7–11, employing, among other things, the trick advised for the periautologia of referring to his own minor defects.[23]

The exhortatory conclusion in verses 15–16 provides a full involvement of the listeners within the Pauline journey passing from "I" to "we". Thus, the Apostle begins to address the Christians of Philippi, considering them τέλειοι, i.e., mature in the faith (cf., e.g., 1 Cor 14:20; Heb 5:14) and therefore called to assume the mentality just shown in Paul's itinerary (τοῦτο φρονῶμεν). If this is the essential perspective to be taken into account, for the rest, it is left exclusively to God to enlighten the listeners, through his revealing in the case of disagreements with Paul on minor issues. Ultimately, according to verse 16, for the Philippians, as for the founder of their church, it is a matter of maintaining the level of Christian life achieved and of moving forward united and unanimous. In these verses, the typical device of the transfer, already implied in verse 3, occurs between the author and the recipients; both are praised as "perfect", even if, paradoxically, this condition, as explained in verse 12, consists of the awareness of their own imperfection in the Christian life.[24]

### 3.5.4. Exhortation with Motivations (3,17–21)

The final textual unit A'. (verses 17–21) is composed of an exhortation to imitate Paul (verse 17) and the two reasons for it (verses 18–19.20–21). In particular, in verse 17, there is a transition from the previous "we" to "you", through a positive invitation, addressed to the recipients, to imitate the Apostle all together,[25] helped also by those who already follow his model. Therefore, this verse indicates not only the purpose of the entire passage of Phil 3:1–4:1 but also the higher, justifying, ethical purpose of the Pauline periautologia; Paul has demonstrated his example for the Christians of Philippi to imitate (and not only them).

In spite of this conclusion, the direct appeal to imitate the speaker in Phil 3:17 could still turn out to be a demonstration of arrogant superiority, since it would represent a unique case in all ancient thought before Paul.[26] Indeed, such an exhortation is understandable and acceptable only in view of the Apostle's unique awareness of the new identity he has received. In the comparison with the past, he sees his own existence radically transformed and expropriated in order to "live Christ" (cf. Phil 1:21) so that he is able to speak of himself as "other-than-self" and propose himself as a model for others as a concrete image of his own Lord.

The first motivation (note γάρ) of the exhortation is negative, and it is the threat posed by the bad example of the enemies (verses 18–19), which is also a typical reason to turn to periautologia. As already stated in verse 2, now, in verse 18, they are denigrated by the author so that the listeners—repeatedly warned by the Apostle and now pleaded with in tears (use of rhetorical *pathos* to indicate urgency)—will not be influenced by them. Indeed, the opponents are described as those whose behaviour is completely at variance with the cross of Christ (cf. 1 Cor 1:18–25).[27] They, in fact, possess a purely worldly mentality (οἱ τὰ ἐπίγεια φρονοῦντες, verse 19) and not the mentality proper to Christians, who have Christ himself as their point of reference (cf. 2:5).[28]

The second motivation (note γάρ) of the exhortation is positive; it depends on the condition of the Philippians and Paul (and of all Christians) who are destined for final salvation (verses 20–21). They, while living their earthly life, are governed[29] from heaven by their Lord, of whom they are fervently waiting as Saviour (verse 20). For he will one

day come to transfigure the poor bodies of believers, marked by weakness and death, to make them conform to his glorious body, through the energy with which the Risen One exercises his universal dominion (verse 21). In the past, the unusual and particularly elevated language has called into question the Pauline origin of verses 20–21 and led to the hypothesis of a hymn or fragment of a hymn; however, the reasons given are not convincing to most current scholars, since from the study of form, vocabulary, context, and ideas, it is only plausible to assume the Pauline use of traditional material in order to compose a prose text of elevated style as an appropriate *climax* to the entire passage (Matta 2013, pp. 332–37).

Finally, we must note that the procedure of periautological transfer from the author to the audience begun in verse 15 and was completed in verses 20–21 with a eulogy of the "we" group and its identity (also placed in a rhetorical comparison with the enemies of verses 18–19). However, in its turn, this boast of the listeners is subjected to another transfer in relation to Christ. Therefore, the path traced by Phil 3:1–4:1 finds its goal in these last verses; Paul's boast of the self, transformed into a eulogy of Christ, becomes the boast of the Philippians, and, in a wider sense, of all the believers. As such, it will be revealed definitively with the return of the Lord, ruler of history and the universe.

3.5.5. Conclusion (4,1)

Corresponding to the transition of 3:1, in 4:1, we find the conclusion of the passage by means of an appeal, in a very affectionate tone (in order to arouse a positive *pathos* towards the Apostle), for the believers in Philippi to remain steadfast and faithful to Christ, in the manner just shown in the Pauline example. Overall, the text of 4:1 is not an exhortation in its own right but rather a summary and final reminder of the previous exhortations, the one to beware of adversaries (3:2) and the one to imitate Paul (3:17).

*3.6. The Text in the Context and the Overall Logic of the Text*

After an examination of the passage in itself, the question is what its significance in its context (section and letter) is. The passage 3:1–4:1 clearly recalls the Christological text of 2:6–11; in continual conformity to the death of Christ, united with the hope of attaining the resurrection, Paul's path is a reproduction of the way of the Lord, humiliated to the death of the cross and, therefore, highly exalted by God. On the other hand, if Christ is the perfect prototype at the foundation of all Christian existence (see also Timothy and Epaphroditus who, in 2:19–30, are presented as people who embody the Christological model), Paul is the imperfect example, still in becoming, whom the addressees have before their eyes. His is a concrete experience of life in Christ, which they are called to emulate according to the characteristics of each.

With a concluding glance, we shall summarise the overall logic of the text, bearing in mind the entirety of Pauline theology, which is also revealed in other, possibly parallel, passages. In conclusion, the sense of the text of Phil 3:1–4:1 is that of addressing the believer with an exhortation to follow the Apostle in making his or her own life a paradoxical praise of self, based on the way of Christ so that it becomes a praise of the Lord. Extending our case to other autobiographical Pauline passages such as 2 Cor 11:1–12:18 and Gal 1:11–2:21, we receive confirmation of the assumption that the Apostle presents the path of his life to the advantage of the Gospel, as also implied summarily by the text of Phil 1:12. In fact, in these texts, the Apostle emphasises the initiative of God who changed his life with a paradoxical and shocking style of human logic; because he chose the one who was proud to be a Jew and was furthest from the gospel to make him a proclaimer for the Gentiles (Gal 1, 11–2:21), this showed him that all his achievements as an observant Pharisee are rubbish in the face of the knowledge of Christ (Phil 3:1–4:1), and this granted him his strength and power when he came to boast of his own weakness (2 Cor 11:1–12:18).

**4. Paul's Way of Speaking about Himself in Phil 3:1–4:1**

Focusing now on Paul's "I" in Phil 3:1–4:1, Vouga's (2000) position is interesting here, because he affirms that the Apostle stands at the end of a path of development of

self-consciousness proper to antiquity, which began with the discovery of the individual in Homeric literature and continued with that of the subject in Greek lyric poetry. In fact, Paul, as a result of the divine revelation he received, becomes a subject capable of reflecting on his "I", a self-reflective subject; this perspective will be further followed and deepened by Augustine in his *Confessiones*, an intimate diary of his soul and the pinnacle of autobiographical writing in antiquity (Cf. Baslez et al. 1993).

In Phil 3:1–4:1, looking back over the entire journey, Paul can put his "I" at the centre, in communication with the recipients, because he now considers himself an "other-than-self". Indeed, he has received a completely new identity through the encounter with the Risen One, and he now possesses, not through his own merit but through a divine gift, an "I in Christ", even though his journey has not yet reached its destination, because he is waiting, together with the other believers, for the final transformation that will come with the resurrection from the dead.[30] From this perspective, the use of the first-person singular, proper to periautologia, does not prove unpleasant as expected in the context of the ancient world, for the Apostle is essentially speaking of someone other than what he was thanks to the gifts he received and the merits acquired. Moreover, the exemplarity of the "I" of Phil 3:1–4:1, with regard to Christian existence, can also be well understood by the Philippians so as to accept the massive use of the first-person singular in the Pauline text (Dodd 1999). After all, the use of the "I" is a suitable and effective tool to enable the process of identification of the recipients with the author so as to lead them precisely to emulate his itinerary (Cf. Schoenborn 1989). It is an itinerary that does not focus on the subject but rather on what God has accomplished in the person. In this sense, talking about oneself becomes the perfect means of highlighting all the greatness and mercy of God, who can radically change a person's life with his grace, and therefore, what could be an excellent occasion for self-praise becomes instead an ideal situation for praising one's Lord. Ultimately, by talking about himself, Paul has found the best way to talk about Christ.

**Funding:** This research received no external funding.

**Data Availability Statement:** No new data were created or analyzed in this study. Data sharing is not applicable to this article.

**Conflicts of Interest:** The author declares no conflict of interest.

## Notes

[1] For a comprehensive study of the autobiographical Pauline texts, we permit ourselves to cite (Bianchini 2021).

[2] The pioneering study was (Betz 1975). For a good survey of the reactions to Betz and the rhetorical analysis studies of the NT till 2009, see (Classen 2009); very useful, though just less updated: (Lampe 2006; Watson 2006). Moreover, Aletti (2021) provides an overview of the rhetorical studies of the Pauline letters up to the present day. He speaks of a first generation (Betz and his followers) with the strict application of the classic model of the forensic rhetoric to the letters; a second generation trying to find only the essential elements of each argumentation in the texts and building a bridge between epistolography and rhetoric; and a third generation that focuses not only on the arrangement of the argumentation but also on its proof and their evaluation. The last interesting contribution about Paul's rhetoric is Thurén (2022, pp. 294–313).

[3] For further development of the history of rhetorical analysis in Paul, from which the following is a summary, see (Bianchini 2023).

[4] (Aletti 1992, 1996). However, the denomination "literary rhetoric" is never used by Aletti, but following this methodology, it is coined by Pitta (1996, pp. 36–37).

[5] This paragraph builds on (Bianchini 2006).

[6] The most recent work that holds this view is (Standhartinger 2021, pp. 20–22).

[7] One of the last contributions that follow this perspective is (Wick 2015, pp. 309–26).

[8] After my book *L'elogio di sé in Cristo* (2006), see (Pitta 2010, pp. 208–11; Smit 2013, pp. 118–21; Focant 2015, p. 144; Gerber 2015; Pialoux 2017, pp. 244–76; Giuliano 2019; Rojas 2019).

[9] For a critical edition of this work, see (Plutarque 1974, pp. 539A–47F).

[10] Plutarch, *On Praising Oneself*, 539E. 545E.

[11] For a good and comprehensive discussion of all the interpretive issues concerning 3:1, see (Aletti 2005).

[12] For a convincing explanation of the words of verse 2, see, e.g., (Williams 2002, pp. 154–59).

[13]  Here, "Law" indicates the Mosaic law.

[14]  For a good state of research about adversaries' identity in Philippians, see (Nikki 2019, pp. 8–22).

[15]  Paul begins with circumcision because, as already appears in the previous verse, it was to be what the adversaries could demand of the recipients, but also because, as Aletti (2005, p. 31) well points out, it is this that constitutes the fundamental religious identity of the Jew, and therefore, the following three privileges without it would be purely worldly for him.

[16]  As Holloway (2017, p. 160) points out, it should be noted that in Phil 3:6, Paul has no fear of arguing for possible perfect obedience to the Law, thus defeating many interpreters who claim that, for the Apostle, the deficiency of the Law would be found in the impossibility of observing it fully. Clearly, here we are in the realm of Paul's past life in Judaism; in fact, furthermore, the righteousness derived from the Law will be contrasted with that based on faith in Christ, which will prove to be the only one capable of justification (3:9).

[17]  The Grek term σκύβαλα utilized here can be also translated as "dung".

[18]  The syntactic structure of these verses is very complex. Bear in mind that in verse 9, καὶ εὑρεθῶ ἐν αὐτῷ is inseparable from ἵνα Χριστὸν κερδήσω, a sentence dependent on ἡγοῦμαι of verse 8. In turn, τοῦ γνῶναι of verse 10 is syntactically dependent on ἡγοῦμαι of verse 8, and it is thus parallel to ἵνα Χριστὸν κερδήσω καὶ εὑρεθῶ ἐν αὐτῷ. Finally, εἴ πως καταντήσω of verse 11 depends on of συμμορφιζόμενος of verse 10.

[19]  For all the exegetical and theological problems raised by the verse, we permit ourselves to refer to (Bianchini 2011).

[20]  A recent valid contribution about the relation between the journey of Christ in Ph 2 and the journey of Paul in Phil 3 is (Bertschmann 2018).

[21]  The text carefully distinguishes the believer's present participation in Christ's resurrection (τῆς ἀναστάσεως αὐτοῦ, verse 10) from the future participation that is from the dead (τὴν ἐξανάστασιν τὴν ἐκ νεκρῶν, verse 11).

[22]  E.g., Seneca Jr., *Moral Epistles to Lucilius* 78.16 and Philo, *Life of Moses* 1.48.

[23]  As Aletti (2005, p. 253) suggests, highlighting the *periautological* logic of these verses, the positions of those who want to see a response to opponents (e.g., perfectionists or with a realised eschatology) in the Pauline statements of verses 12–14 are unfounded.

[24]  From a rhetorical point of view, the pair consisting of the negation of τετελείωμαι in verse 12 and the affirmation of being τέλειοι in verse 15 is an antanaclasis, i.e., a repetition of the same word (or body of words) in two different senses.

[25]  The word συμμιμητής is *hapax legomenon* in all Greek literature up to that time. The simplex form μιμητής is used in the Pauline letters for the imitation of other communities (1 Thess 2:14), of God (Eph 5:1), of Christ and Paul together with his co-workers (1 Cor 11:1; 1 Thess 1:6), and of the Apostle alone (1 Cor 4:16).

[26]  We do not know, however, whether the invitation to imitate in 4 Mac 9:23 is earlier or later.

[27]  In verse 18, we have a very complicated syntactic–grammatical structure; in this regard, we can note the hyperbaton because the parenthetic phrase (οὓς πολλάκις ἔλεγον ὑμῖν, νῦν δὲ καὶ κλαίων λέγω) is interposed between two constituents of a syntagma in order to highlight the expression τοὺς ἐχθροὺς τοῦ σταυροῦ τοῦ Χριστοῦ.

[28]  The text does not indicate the involvement of a new group of opponents; however, Paul intends to broaden the discourse, since the elements in verses 18–19 do not only apply to Judeo-Christian opponents but are aimed at blaming all those who lead an existence in contradiction with the gospel of the cross of Christ, who, in different ways, could influence the recipients.

[29]  The word πολίτευμα (in the New Testament, *hapax legomenon*), denotes the result or dynamic of the action expressed by the verb πολιτεύω, used in 1:27, and thus possesses a basic sense of "political activity". There are four proposals regarding the meaning of the term in 3:20 (cf. Aletti 2005, pp. 273–75): "citizenship", "colony", "homeland", "state, constitution". The last meaning is the one most frequently witnessed in the Hellenistic period so as to indicate, in our context, the model and the force that governs the earthly life of believers; this reality is found in the heavens, and is therefore placed, as the end of the verse suggests, in direct connection with Christ himself.

[30]  Becker (2019) speaks of an introspective Pauline "I" in Phil 1–3, which, as is the case with ancient philosophers (see especially Seneca), develops its reflection above all in view of imminent death; furthermore, in Phil 3, there is an eschatological transformation of Paul's "I". For our part, we find the German scholar's contribution interesting. However, we believe that the prospect of imminent death is only present in Phil 1 and that the transformation of the Apostle's "I" in Phil 3 is first and foremost with respect to his Jewish pharisaic identity in the past.

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
