# Peer review of "Talking about Oneself to Talk about Christ: The Autobiographical Text of Philippians 3:1–4.1 in Light of Ancient Rhetorical Heritage"

_religions, doi:10.3390/rel15040398_

Round 1

Reviewer 1 Report

Comments and Suggestions for Authors

Comments on the Quality of English Language

There were a few typos and minor grammatical infelicities that I noted while reading the manuscript. I have placed them in a table in the file that is attached to this review. Since the table  is not intended to be exhaustive, a fuller proofreading will be useful before publication.

Author Response

Thank you for your review that I appreciated.

Reviewer 2 Report

Comments and Suggestions for Authors

The proposed article contributes to the body of knowledge on rhetorical approaches in the New Testament as well as to our understanding of how Paul applies rhetorical techniques.

I bit more engagement with the various perspectives such as the so-called "New Perspective(s) on Paul" is expected in places in which justification and the law is at play. Some statements seem to be with unawareness of these perspectives, e.g., in lines 326-329: "Thus, in the verse, the opposition is between «Law» and «faith in Christ» as two contrary and alterna tive principles on which to base one's righteousness, where only the last one leads to right eousness derived from God and therefore salvific. Paul, in his choice for Christ, opted for the second principle and abandoned the first"  Proponents of the NPP would not necessarily place faith and the law on opposite ends, for example. The author should show more awareness of the NPP approaches in general.

Author Response

Thank you for your review. I am not neither a NPP scholar nor a Lutheran Paul scholar. The aforementioned is my interpretation of the text of Phil 3.  This position of mine is explained in detail in the article I quoted in the footnote.

Reviewer 3 Report

Comments and Suggestions for Authors

It isn't easy to comment on the present format of the paper.    According to the journal format, and also a common understanding of abstract for a journal article, "The abstract should be a total of about 200 words maximum."    (https://www.mdpi.com/journal/religions/instructions#front However, the length of the present version's abstract is more than one page.  I need a precise summary of the method and thesis in the abstract (less than 200 words) as a map to be a point of departure for my evaluation of the paper.

Comments on the Quality of English Language

Better not use so many times "First of all" to avoid confusion.

Author Response

Thank you for your review. I am very sorry for the length of the abstract, but according to the rules coming from Dr. Blois this was the length. Nevertheless, now the abstract contains less then 200 words.

Round 2

Reviewer 3 Report

Comments and Suggestions for Authors

Author Response

Thank you for your review. I followed all your indications, except the third because in note 25 I explain the syntactical structure of verses 7-11 and I think that is enough for the reader.